# Comparing Distributions by Measuring Differences that Affect Decision Making

**Shengjia Zhao**\*, **Abhishek Sinha**\*, **Yutong He**\*, **Aidan Perreault, Jiaming Song, Stefano Ermon**
Department of Computer Science
Stanford University
`{sjzhao,a7b23,kellyyhe,aperr,tsong,ermon}@stanford.edu`

## Abstract

Measuring the discrepancy between two probability distributions is a fundamental problem in machine learning and statistics. We propose a new class of discrepancies based on the optimal loss for a decision task – two distributions are different if the optimal decision loss is higher on their mixture than on each individual distribution. By suitably choosing the decision task, this generalizes the Jensen-Shannon divergence and the maximum mean discrepancy family. We apply our approach to two-sample tests, and on various benchmarks, we achieve superior test power compared to competing methods. In addition, a modeler can directly specify their preferences when comparing distributions through the decision loss. We apply this property to understanding the effects of climate change on different economic activities and selecting features targeting different decision tasks.

## 1 Introduction

Quantifying the difference between two probability distributions is a fundamental problem in machine learning. Modelers choose different types of discrepancies (or probability divergences) to encode their prior knowledge about which aspects are relevant to evaluate the difference. Integral probability metrics (IPMs, Müller (1997)) and $f$-divergences (Csiszár, 1964) are widely used discrepancies in machine learning. IPMs, such as the Wasserstein distance, maximum mean discrepancy (MMD) (Rao, 1982; Burbea & Rao, 1984; Gretton et al., 2012), are based on the idea that if two distributions are identical, any function should have the same expectation under both distributions. IPMs are used to define training objectives for generative models (Arjovsky et al., 2017), perform independence tests (Doran et al., 2014), robust optimization (Esfahani & Kuhn, 2018) among many other applications. $f$-divergences, such as the KL divergence and the Jensen Shannon divergence, are based on the idea that if two distributions are identical, they assign the same likelihood to every point. One can then define a discrepancy based on how different the likelihood ratio is from one. KL divergence underlies some of the most commonly used training objectives for both supervised and unsupervised machine learning algorithms, such as cross entropy loss.

We propose a third category of divergences called H-divergences that overlaps with but also extends the set of integral probability metrics or the set $f$-divergences. Intuitively, H-divergence compares two distributions in terms of the optimal loss for a certain decision task. This optimal loss corresponds to a generalized notion of entropy (DeGroot et al., 1962). Instead of measuring the best average code length of any encoding scheme (Shannon entropy), the generalized entropy uses arbitrary loss function (rather than code length) and set of actions (rather than encoding schemes), and is defined as the best expected loss among the set of actions. In particular, given two distribution $p$ and $q$, we compare the generalized entropy of the mixture distribution $(p+q)/2$ and the generalized entropy of $p$ and $q$ individually. Intuitively, if $p$ and $q$ are different, it is more difficult to minimize expected loss under the mixture distribution $(p+q)/2$, and hence the mixture distribution should have higher generalized entropy; if $p$ and $q$ are identical, then the mixture distribution is identical to $p$ or $q$, and hence should have the same generalized entropy.

Our divergence strictly generalizes the maximum mean discrepancy family and the Jensen Shannon divergence, which can be obtained with specific choices of the loss function. We illustrate this via

---

\*Co-first author

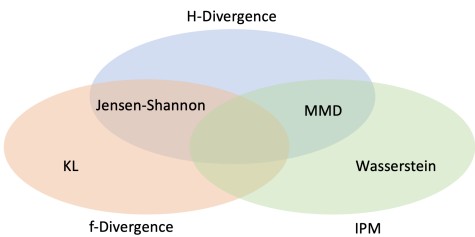

Figure 1: Relationship between H-divergence (this paper) and existing divergences. The Jensen Shannon divergence is an $f$-divergence but not an IPM; the MMD is an IPM but not always an $f$-divergence; both are H-divergences. There are H-divergences that are not $f$-divergences or IPMs.

the Venn diagram in Figure 1. Our formulation allows us to choose alternative losses to leverage inductive biases and machine learning models from different problem domains. For example, if we choose the generalized entropy as the maximum log likelihood of deep generative models, we are able to leverage recent progress in modeling high dimensional images.

We demonstrate the effectiveness of H-divergence in two sample tests, i.e. to decide whether two sets of samples come from the same distribution or not. A test based on a probability discrepancy declares two sets of samples different if their discrepancy exceeds some threshold. We use H-divergences based on generalized entropy defined by the log likelihood of off-the-shelf generative models. Compared to state-of-the-art tests based on MMD with deep kernels (Liu et al., 2020), tests based on the H-divergence achieve better test power (given identical type I error) on a large set of benchmarks.

More importantly, scientists and policy makers are often interested not only in **if** two distributions are different, but **how** two distributions are different and whether the differences affect decision making. Typical divergence measures (such as KL) or two sample tests only quantify if two distributions are different, while we show that H-divergence is a useful tool for quantifying how distributions are different with three application examples: studying the effect of climate change, feature selection, and sample quality evaluation. In each of these examples, we compare different aspects of the distributions by choosing specific decision loss functions. For example, climate change (Figure 3) might impact agriculture in a region but not energy production, or vice versa. By choosing suitable loss functions (related to agriculture, energy, etc) we can quantify and test if the change in climate distribution impact different economic activities.

## 2 BACKGROUND

### 2.1 PROBABILITY DIVERGENCES

Let $\mathcal{X}$ denote a finite set or a finite dimensional vector space, and $\mathcal{P}(\mathcal{X})$ denote the set of probability distributions on $\mathcal{X}$ that have a density. We consider the problem of defining a probability divergence between any two distributions in $\mathcal{P}(\mathcal{X})$, where a probability divergence is any function $D : \mathcal{P}(\mathcal{X}) \times \mathcal{P}(\mathcal{X}) \to \mathbb{R}$ that satisfies $D(p\|q) \geq 0, D(p\|p) = 0, \forall p, q \in \mathcal{P}(\mathcal{X})$. We call the divergence $D$ "strict" if $D(p\|q) > 0 \; \forall p \neq q$, and "non-strict" otherwise. In this paper we consider both types of divergences.

**Integral Probability Metrics** Let $\mathcal{F}$ denote a set of functions $\mathcal{X} \to \mathbb{R}$. An integral probability metric is defined as $\text{IPM}_{\mathcal{F}}(p\|q) = \sup_{f \in \mathcal{F}} |\mathbb{E}_p[f(X)] - \mathbb{E}_q[f(X)]|$. Several important divergences belong to integral probability metrics. Examples include the Wasserstein distance, where $\mathcal{F}$ is the set of 1-Lipschitz functions; the total variation distance, where $\mathcal{F}$ is the set of functions $\mathcal{X} \to [-1, 1]$. The maximum mean discrepancy (MMD) (Rao, 1982; Burbea & Rao, 1984; Gretton et al., 2012) chooses a kernel function $k : \mathcal{X} \times \mathcal{X} \to \mathbb{R}_+$ and is defined by

$$\text{MMD}^2(p\|q) = \mathbb{E}_{p,p} k(X, Y) + \mathbb{E}_{q,q} k(X, Y) - 2\mathbb{E}_{p,q} k(X, Y)$$

MMD is an IPM where $\mathcal{F}$ is the unit norm functions in the reproducing kernel Hilbert space (RKHS) associated with the kernel $k$.

$f$**-Divergences** Given any convex continuous function $f : \mathbb{R}_+ \to \mathbb{R}$ such that $f(1) = 0$, the $f$-Divergence is defined as (assuming densities exist) $D_f(p\|q) = \mathbb{E}_q[f(p(X)/q(X))]$. Examples include the KL divergence, where $f : t \mapsto t \log t$ and the Jensen Shannon divergence, where $f : t \mapsto (t+1)\log\left(\frac{2}{t+1}\right) + t \log t$.

## 2.2 H-ENTROPY

For any action space $\mathcal{A}$ and any loss function $\ell : \mathcal{X} \times \mathcal{A} \to \mathbb{R}$, the H-entropy (DeGroot et al., 1962; DeGroot, 2005; Grünwald et al., 2004) is defined as

$$H_\ell(p) = \inf_{a \in \mathcal{A}} \mathbb{E}_p[\ell(X, a)]$$

In words, $H$-entropy is the Bayes optimal loss of a decision maker who must select some action $a$ not for a particular $x$, but in expectation for a random $x$ drawn from $p(x)$. H-entropy generalizes several important notions of uncertainty. Examples include: *Shannon Entropy*, where $\mathcal{A}$ as the set of probabilities $\mathcal{P}(\mathcal{X})$, and $\ell(x, a) = -\log a(x)$; *Variance* where $\mathcal{A} = \mathcal{X}$, and $\ell(x, a) = \|x - a\|_2^2$; *Predictive $\mathcal{V}$-entropy*, where $\mathcal{A} \subset \mathcal{P}(\mathcal{X})$ is some subset of distributions, and $\ell(x, a) = -\log a(x)$ (Xu et al., 2020).

A key property we will use is that $H$-entropy is concave (DeGroot et al., 1962).

**Lemma 1.** *For any choice of $\ell : \mathcal{X} \times \mathcal{A} \to \mathbb{R}$, $H_\ell$ is a concave function.*

This Lemma can be proved by observing that $\inf$ is a concave function: it is always better to pick an optimal action for $p$ and $q$ separately rather than a single one for both.

$$H_\ell\big(\alpha p + (1-\alpha)q\big) = \inf_a \left(\alpha \mathbb{E}_p[\ell(X, a)] + (1-\alpha)\mathbb{E}_q[\ell(X, a)]\right)$$
$$\geq \alpha \inf_a \mathbb{E}_p[\ell(X, a)] + (1-\alpha)\inf_a \mathbb{E}_q[\ell(X, a)] = \alpha H_\ell(p) + (1-\alpha)H_\ell(q)$$

This Lemma reflects why $H_\ell$ can be thought of as a measurement of entropy or uncertainty. If the distribution is more uncertain (e.g. a mixture of $p$ and $q$, rather than $p$ or $q$ separately) then decisions made under higher uncertainty will suffer a higher loss.

## 3 DEFINITION AND THEORETICAL PROPERTIES

### 3.1 H-JENSEN SHANNON DIVERGENCE

As a warm up, we present a special case of our divergence.

**Definition 1** (H-Jensen Shannon divergence)**.**

$$D_\ell^{\text{JS}}(p, q) = H_\ell\left(\frac{p+q}{2}\right) - \frac{1}{2}\big(H_\ell(p) + H_\ell(q)\big) \tag{1}$$

$D_\ell^{\text{JS}}$ is always non-negative because H-entropy is concave (Lemma 1), and clearly $D_\ell^{\text{JS}}(p, q) = 0$ whenever $p = q$. Therefore, $D_\ell^{\text{JS}}$ is a valid probability divergence. In particular, if we choose $H_\ell$ as the Shannon entropy, Definition 1 recovers the Jensen Shannon divergence. Other special loss function choices can recover definitions in (Burbea & Rao, 1982).

### 3.2 GENERAL H-DIVERGENCE

In addition to the H-Jensen Shannon divergence, there are other functions based on the H-entropy that satisfy the requirements of a divergence. For example,

$$D_\ell^{\text{Min}} = H_\ell\left(\frac{p+q}{2}\right) - \min(H_\ell(p), H_\ell(q)) \tag{2}$$

is also a valid divergence (this will be proved later as a special case of Lemma 2). We can define a general set of divergences that includes the above two divergences with the following definition:

**Definition 2** (H-divergence). *For two distributions $p, q$ on $\mathcal{X}$, given any continuous function $\phi :$ $\mathbb{R}^2 \to \mathbb{R}$ such that $\phi(\theta, \lambda) > 0$ whenever $\theta + \lambda > 0$ and $\phi(0, 0) = 0$, define*

$$D_\ell^\phi(p\|q) = \phi\left(H_\ell\left(\frac{p+q}{2}\right) - H_\ell(p), H_\ell\left(\frac{p+q}{2}\right) - H_\ell(q)\right)$$

Intuitively $H_\ell\left(\frac{p+q}{2}\right) - H_\ell(p)$ and $H_\ell\left(\frac{p+q}{2}\right) - H_\ell(q)$ measure how much more difficult it is to minimize loss on the mixture distribution $(p+q)/2$ than on $p$ and $q$ respectively. $\phi$ is a general class of functions that map these differences into a scalar divergence, while satisfying some desirable properties described in the next section.

The following proposition shows that the H-divergence generalizes the previous definitions (1) and (2). Therefore, any property of H-divergence is inherited by e.g. the H-Jensen Shannon divergence.

**Proposition 1.** *If $\phi(\theta, \lambda) = \frac{\theta + \lambda}{2}$ then $D_\ell^\phi(p, q)$ is the H-Jensen Shannon divergence in Eq.(1). If $\phi(\theta, \lambda) = \max(\theta, \lambda)$ then $D_\ell^\phi(p, q)$ is the H-Min divergence in Eq.(2).*

### 3.3 PROPERTIES OF THE H-DIVERGENCE

We first verify that $D_\ell^\phi$ is indeed a (strict or non-strict) probability divergence.

**Lemma 2.** *For any choice of $\ell$ and for any choice of $\phi$ that satisfy Definition 2, $D_\ell^\phi$ is non-negative and $D_\ell^\phi(p, q) = 0$ whenever $p = q$. Furthermore, $D_\ell^\phi$ is symmetric whenever $\phi$ is symmetric.*

Depending on the choice of $\ell$, H-divergence may or may not be strict (i.e. whenever $p \neq q$, $D(p\|q) > 0$). The following proposition characterizes conditions for a strict divergence.

**Proposition 2** (Strict Divergence). *For any choice of $\phi$ the following are equivalent 1) $\forall p \neq q$, $D_\ell(p\|q) > 0$. 2) The H-entropy $H_\ell(p) := \inf_a \mathbb{E}_p[\ell(X, a)]$ is strictly convex in $p$. 3) $\forall p \neq q$, $\arg\inf_a \mathbb{E}_p[\ell(X, a)] \cap \arg\inf_a \mathbb{E}_q[\ell(X, a)] = \emptyset$.*

In particular, this proposition can be used to characterize *all* strict H-divergences, because the set of all losses $\ell$ that induces strict H-entropy functions $H_\ell$ can be characterized by Fenchel duality (Duchi et al., 2018).

One important property of the H-divergence is that two distributions have non-zero divergence if and only if they have different optimal actions, i.e. the optimal solutions for their respective H-entropy are different. This is shown in the following proposition (proof in Appendix A).

**Proposition 3.** *$D_\ell^\phi(p\|q) > 0$ if and only if $\arg\inf_a \mathbb{E}_p[\ell(X, a)] \cap \arg\inf_a \mathbb{E}_q[\ell(X, a)] = \emptyset$.*

Intuitively, $D_\ell^\phi$ only takes into account differences between distributions that lead to different optimal action choices. This property allows us to incorporate prior domain knowledge. By choosing $\mathcal{A}$ and $\ell$ we can specify which differences between distributions lead to different optimal actions, and which differences do not. For example, we can choose $\mathcal{A}$ as a set of generative models (e.g., mixture of Gaussians) and $\ell(x, a)$ as the negative log likelihood of $x$ under generative model $a$. If under two distributions we end up learning the same generative model (by maximizing log likelihood), the H-divergence between them is zero.

### 3.4 RELATIONSHIP TO MMD

An important special case of the H-divergence is the set of squared Maximum Mean Discrepencies (MMD), as shown by the following theorem:

**Theorem 1.** *The set of H-Jensen Shannon Divergences is strictly larger than the $\mathrm{MMD}^2$ distances.*

To prove this theorem, we show that for each choice of kernel $k : \mathcal{X} \times \mathcal{X} \to \mathbb{R}$, there exists an action space $\mathcal{A}$ and loss $\ell$ such that the corresponding squared MMD distance and H-divergence are the same (see proof in Appendix A). In particular, this equivalence can be achieved by choosing $\mathcal{A}$ to be the RKHS $\mathcal{H}$ of $k(\cdot, \cdot)$, and $\ell(x, a) = 4\|k(x, \cdot) - a\|_\mathcal{H}^2$. Inclusion is strict because the Jensen Shannon divergence is a H-Jensen Shannon Divergence but not a squared MMD distance.

### 3.5 ESTIMATION AND CONVERGENCE

Many machine learning tasks can be reduced to the problem of estimating the divergence between two distributions given samples. Specifically, suppose we are provided with a set of $m$ i.i.d. samples $\hat{p}_m = (x_1, \cdots, x_m)$ drawn from distribution $p$ and $\hat{q}_m = (x'_1, \cdots, x'_m)$ drawn from distribution $q$, and would like to obtain an estimate of $D_\ell^\phi(p\|q)$ based on the samples. Here, $\hat{p}_m$ and $\hat{q}_m$ denote empirical distributions drawn from $p$ and $q$ respectively. In this section we propose an empirical estimator for the H-divergence and show that it has favorable convergence properties.

Let $\hat{D}_\ell^\phi(\hat{p}_m\|\hat{q}_m)$ be the empirical (random) estimator for $D_\ell^\phi(p\|q)$ defined by

$$\hat{D}_\ell^\phi(\hat{p}_m\|\hat{q}_m) = \phi\left(\inf_a \frac{1}{m}\sum_{i=1}^m \ell(x_i'', a) - \inf_a \frac{1}{m}\sum_{i=1}^m \ell(x_i, a), \inf_a \frac{1}{m}\sum_{i=1}^m \ell(x_i'', a) - \inf_a \frac{1}{m}\sum_{i=1}^m \ell(x_i', a)\right)$$

where $x_i'' = x_i b_i + x_i'(1 - b_i)$ and $b_i$ are i.i.d uniformly sampled from $\{0, 1\}$, so that $x_i''$ is a sample from the mixture distribution $(p + q)/2$ of size $m$.

Using $x_i''$ as defined above is crucial for the convergence properties we will prove in Theorem 2. It might be tempting to replace the term $\frac{1}{m}\sum_{i=1}^m \ell(x_i'', a)$ with $\frac{1}{2m}\sum_{i=1}^m (\ell(x_i, a) + \ell(x_i', a))$ to use all the available samples. However, optimizing the action based on a finite set of samples (instead of in expectation) is prone to overfitting, and introduces bias. Intuitively, using $m$ samples ($x_i''$) ensures the bias for the mixture is comparable to that of $p$ and $q$. Without this, Theorem 2 is no longer true, and empirical performance also degrades.

Before presenting the convergence results, we first must define several assumptions that make convergence possible. In particular, we are going to assume that the loss function $\ell$ is $C$-bounded, i.e. there exists some $C$ such that $0 \le \ell(x, a) \le C, \forall a, x$. This assumption seemingly exclude important special cases such as the Jensen-Shannon divergence (which is associated with the unbounded log loss). However, we show in the appendix that the Jensen-Shannon divergence cannot be consistently estimated in general, hence correctly excluded by our theorem. One practical solution is to clip the log likelihood, which is the approach adopted in (Song & Ermon, 2019) for improved divergence estimation (for a similar KL divergence estimation problem).

In addition, we assume that $\phi$ is 1-Lipschitz under the $\infty$-norm, i.e. $|\phi(\theta + d\theta, \lambda + d\lambda) - \phi(\theta, \lambda)| \le \max(d\theta, d\lambda), \forall \theta, \lambda, d\theta, d\lambda \in \mathbb{R}$. Both $\phi(\theta, \lambda) = \frac{\theta + \lambda}{2}$ and $\phi(\theta, \lambda) = \max\{\theta + \lambda\}$ are 1-Lipschitz under the $\infty$-norm. This is a mild assumption because if $\phi$ is not 1-Lipschitz we can rescale $\phi$ to make it 1-Lipschitz. Finally, define the Radamacher complexity

$$\mathcal{R}_m^p(\ell) = \mathbb{E}_{X_i \sim p, \epsilon_i \sim \text{Uniform}(\{-1,1\})}\left[\sup_{a \in \mathcal{A}} \frac{1}{m}\sum_{i=1}^m \epsilon_i \ell(X_i, a)\right]$$

We define $\mathcal{R}_m^q(\ell)$ analogously. Based on these assumptions and definitions we can bound the difference between $\hat{D}_\ell^\phi(\hat{p}_m\|\hat{q}_m)$ and $D_\ell^\phi(p\|q)$.

**Theorem 2.** *If $\ell$ is $C$-bounded, and $\phi$ is 1-Lipschitz under the $\infty$-norm, for any choice of distribution $p, q \in \mathcal{P}(\mathcal{X})$ and $t > 0$ we have*

1. $\Pr[\hat{D}_\ell^\phi(\hat{p}_m\|\hat{q}_m) \ge t] \le 4e^{-\frac{t^2 m}{2C^2}}$ *if $p = q$.*

2. $\Pr\left[\left|\hat{D}_\ell^\phi(\hat{p}_m\|\hat{q}_m) - D_\ell^\phi(p\|q)\right| \ge 4\max(\mathcal{R}_m^p(\ell), \mathcal{R}_m^q(\ell)) + t\right] \le 4e^{-\frac{t^2 m}{2C^2}}$

**Corollary 1.** $\sqrt{\text{Var}[\hat{D}_\ell^\phi(\hat{p}_m\|\hat{q}_m)]} \le 4\max(\mathcal{R}_m^p(\ell), \mathcal{R}_m^q(\ell)) + \sqrt{2C^2/m}$

For proof see Appendix A. Note that when $p = q$, the convergence of $\hat{D}_\ell^\phi(\hat{p}_m\|\hat{q}_m)$ does not depend on the Radamacher complexity of $\ell$, and converges to 0 very quickly. When $p \ne q$ the estimator $\hat{D}_\ell^\phi(\hat{p}_m\|\hat{q}_m)$ is still consistent (under regularity assumptions)

**Corollary 2.** *[Consistency] Under the condition of Theorem 2, if additionally either 1. $\mathcal{A}$ is a finite set 2. $\mathcal{A}$ is a bounded subset of $\mathbb{R}^d$ for some $d \in \mathbb{N}$ and $\ell$ is Lipschitz w.r.t. $a$, then almost surely $\lim_{m\to\infty} \hat{D}_\ell^\phi(\hat{p}_m\|\hat{q}_m) = D_\ell^\phi(p\|q)$.*

For both cases in Corollary 2 the Radamacher complexity $\mathcal{R}_m^p(\ell)$ goes to zero (as sample size $m \to \infty$) at a rate of $O(1/\sqrt{m})$. In other words we can conclude that the estimation error in Theorem 2 is bounded by $O(1/\sqrt{m})$ and the variance of the estimator is also bounded by $O(1/\sqrt{m})$ when the sample size $m \to \infty$.

## 4 EXPERIMENT: TWO SAMPLE TEST

The first application is to design more powerful two sample tests. We aim to show that H-divergence allow us to leverage inductive biases for each data type (e.g. image, bio, text) by choosing suitable actions $\mathcal{A}$ and loss $\ell$, which leads to improved test power. [1]

### 4.1 TWO SAMPLE TEST

For the task of two sample test, we would like to decide if two sets of samples are drawn from the same distribution or not. Specifically, given two sets of samples $\hat{p}_m := (x_1, \cdots, x_m) \overset{\text{i.i.d.}}{\sim} p$ and $\hat{q}_m := (x_1', \cdots, x_m') \overset{\text{i.i.d.}}{\sim} q$ we would like to decide if $p = q$. Typical approaches estimate a divergence $\hat{D}(\hat{p}_m \| \hat{q}_m)$ and output $p \neq q$ if the divergence exceeds some threshold.

There are two types of errors: *Type I error* happens when the algorithm incorrectly outputs $p \neq q$; the probability of type I errors is called the significance level. *Type II error* happens when the algorithm incorrectly outputs $p = q$; the probability of *not* making a *Type II error* is called the *test power* (higher is better). Note that both the significance level and the test power are relative to distributions $p$ and $q$.

We follow the typical setup where we guarantee a certain significance level while empirically measuring the test power. In particular, the significance level can be guaranteed with a permutation test (Ernst et al., 2004). In a permutation test, in addition to the original set of samples $\hat{p}_m$ and $\hat{q}_m$, we also uniformly randomly swap elements between $\hat{p}_m$ and $\hat{q}_m$, and sample multiple randomly swapped datasets $(\hat{p}_m^1, \hat{q}_m^1), (\hat{p}_m^2, \hat{q}_m^2), \cdots$. The testing algorithm outputs $p \neq q$ if $\hat{D}(\hat{p}_m \| \hat{q}_m)$ is in the top $\alpha$-quantile among $\{\hat{D}(\hat{p}_m^1 \| \hat{q}_m^1), \hat{D}(\hat{p}_m^2 \| \hat{q}_m^2), \cdots\}$. Permutation test guarantees the significance level (i.e. low Type I error) because if $p = q$ then swapping elements between $\hat{p}_m$ and $\hat{q}_m$ should not change its distribution, so each pair $(\hat{p}_m, \hat{q}_m), (\hat{p}_m^1, \hat{q}_m^1), \cdots$ should have the same distribution. Therefore, $\hat{D}(\hat{p}_m \| \hat{q}_m)$ happens to be in the top $\alpha$-quantile with at most $\alpha$ probability. Note that the significance level guarantee does not rely on accurate estimation of H-divergence in Theorem 2 (accurate H-divergence estimation is still important because the test power does depends on it).

When the choice of $D^\ell$ is not a strict divergence (See Proposition 2) we may falsely conclude $p = q$ ($D(p\|q) = 0$) when in reality $p \neq q$. This is true but inconsequential in *finite data* scenarios. With finite data, it is generally impossible to guarantee the test power (i.e. bounding the probability of concluding $p = q$ when in reality $p \neq q$ for any $p, q$) and prior literature do not provide such guarantees. Hence our guarantee is no weaker than prior two sample test literature.

### 4.2 EXPERIMENT SETUP

**Baselines** We compare our proposed approach with six other divergences. All methods are based on the permutation test explained in Section 4.1. MMD-D (Liu et al., 2020) measures the MMD distance with a deep kernel, while MMD-O (Gretton et al., 2012) measures the MMD distance with a Gaussian kernel. Mean embedding (ME) and smoothed characteristic functions (SCF) (Chwialkowski et al., 2015; Jitkrittum et al., 2016) are distances based on the difference in Gaussian kernel mean embedding at a set of optimized points, or a set of optimized frequencies. C2STS-S & C2ST-L (Lopez-Paz & Oquab, 2017; Cheng & Cloninger, 2019) use a classifier's accuracy distinguishing between the two distributions.

**Comparison Metrics and Setup** All methods have the same significance level (which is provably equal to $\alpha = 0.05$ because of the permutation test), therefore we only consider the test power. We follow Liu et al. (2020) and consider four datasets: Blob (Liu et al., 2020), HDGM (Liu et al.,

---

[1]The code to reproduce our experiments can be found here.

2020), HIGGS (Adam-Bourdarios et al., 2014) and MNIST (LeCun & Cortes, 2010). Our method and all the baseline methods have hyper-parameters. To ensure fair comparison, we follow the same evaluation setup as (Liu et al., 2020) for all methods. We split each dataset into two equal partitions: a training set to tune hyper-parameters, and a validation set to compute the final test output.

**Implementation Details** We choose $\phi(\theta, \lambda) = \left(\frac{\theta^s + \lambda^s}{2}\right)^{1/s}$ for $s > 1$ (which includes the H-Jensen Shannon divergence when $s = 1$ and the H-Min divergence when $s = \infty$). We define $l(x, a)$ as the negative log likelihood of $x$ under distribution $a$, where $a$ is in a certain model family $\mathcal{A}$. We experiment with mixture of Gaussian distributions, Parzen density estimtor and Variational Autoencoder (Kingma & Welling, 2013). Our hyper-parameters consist of the best parameter $s$ and also the best generative model family. Choosing these hyper-parameters might seem cumbersome, but compared to the second best baseline (MMD-D which chooses thousands of deep kernel parameters), we have much fewer hyper-parameters.

We use $\alpha = 0.05$ in all two-sample test experiments. Each permutation test uses 100 permutations, and we run each test 100 times to compute the test power (i.e. the percent of times it correctly outputs $p \neq q$). Finally we plot and report the performance standard deviation by repeating the entire experiment 10 times.

## 4.3 EXPERIMENT RESULTS

The average test powers are reported in Figure 4, Figure 2, Table 1 and Table 3. *Our approach achieves superior test power across the board.* Notably on Higgs we achieve the same test power with 2x fewer samples than the second best test, and on MNIST we can achieve perfect test power even on the smallest sample size evaluated in (Liu et al., 2020).

Following (Liu et al., 2020) we also evaluate the test power as the dimension of the problem increases (Figure 2). *Our test power decreases gracefully as the dimension of the problem increases.* We hypothesize that the test power improvements come from leveraging progress in generative model research: for each type of data (e.g. bio, image, text) there has been decades of research finding suitable generative models; we use commonly used generative models (in modern literature) for each data type (e.g. KDE for low dimensional physics/bio data, VAE for simple images).

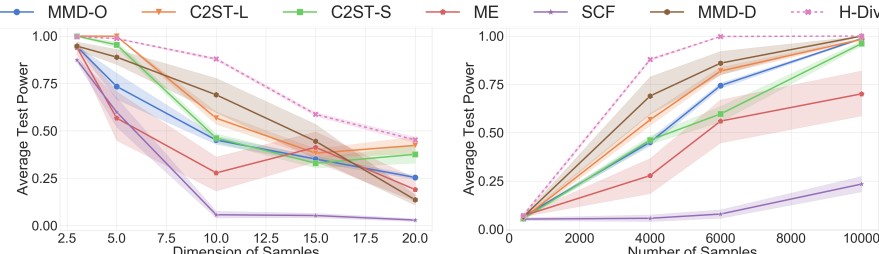

Figure 2: Average test power on HDGM dataset. **Left:** results with the same sample size (4000) and different data dimensions. **Right:** results with the same sample dimension (10) and different sample sizes. Our method (H-Div, dashed line) achieve better test power for almost every setup. All tests have high test power for low data dimensions, but our method scales better for higher data dimensions.

| $N$ | ME | SCF | C2ST-S | C2ST-L | MMD-O | MMD-D | H-Div |
|---|---|---|---|---|---|---|---|
| 1000 | $0.120\pm0.007$ | $0.095\pm0.007$ | $0.082\pm0.015$ | $0.097\pm0.014$ | $0.132\pm0.005$ | $0.113\pm0.013$ | $\mathbf{0.240}\pm\mathbf{0.020}$ |
| 2000 | $0.165\pm0.019$ | $0.130\pm0.019$ | $0.183\pm0.026$ | $0.232\pm0.032$ | $0.291\pm0.017$ | $0.304\pm0.012$ | $\mathbf{0.380}\pm\mathbf{0.040}$ |
| 3000 | $0.197\pm0.012$ | $0.142\pm0.025$ | $0.257\pm0.049$ | $0.399\pm0.058$ | $0.376\pm0.022$ | $0.403\pm0.050$ | $\mathbf{0.685}\pm\mathbf{0.015}$ |
| 5000 | $0.410\pm0.041$ | $0.261\pm0.044$ | $0.592\pm0.037$ | $0.447\pm0.045$ | $0.659\pm0.018$ | $0.699\pm0.047$ | $\mathbf{0.930}\pm\mathbf{0.010}$ |
| 8000 | $0.691\pm0.067$ | $0.467\pm0.038$ | $0.892\pm0.029$ | $0.878\pm0.020$ | $0.923\pm0.013$ | $0.952\pm0.024$ | $\mathbf{1.000}\pm\mathbf{0.000}$ |
| 10000 | $0.786\pm0.041$ | $0.603\pm0.066$ | $0.974\pm0.007$ | $0.985\pm0.005$ | $1.000\pm0.000$ | $1.000\pm0.000$ | $\mathbf{1.000}\pm\mathbf{0.000}$ |
| Avg. | 0.395 | 0.283 | 0.497 | 0.506 | 0.564 | 0.579 | **0.847** |

Table 1: Average test power $\pm$ standard error for $N$ samples over the HIGGS dataset. The results on MNIST is similar and presented in Table 3, Appendix B.1.

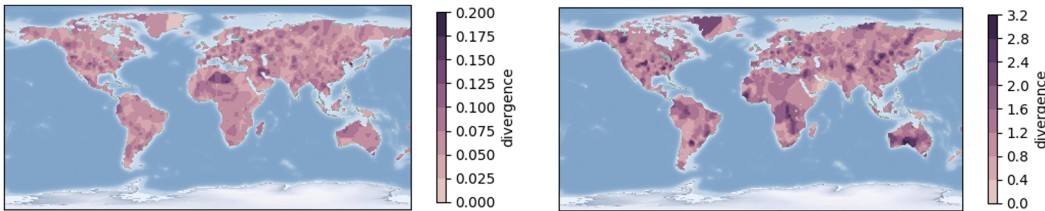

Figure 3: Example plots of H-divergence across different geographical locations for losses $\ell$ related to agriculture (left) and energy production (right). Darker color indicates larger H-divergence. Compared to divergences such as KL, H-divergence measures changes relevant to different social and economic activities (by selecting appropriate loss functions $\ell$). For example, even though climate change significantly impact the high latitude or high altitude areas, this change has less relevance to agriculture (because few agriculture activities are possible in these areas).

## 5 EXPERIMENT: DECISION DEPENDENT DISCREPANCY MEASUREMENT

### 5.1 ASSESSING CLIMATE CHANGE

As an illustrative example of how H-divergence can facilitate decision making, we use climate data and study how climate change affects decision making through the lens of H-divergence. Scientists and policy makers are often interested in how climate change disparately affect different geographical locations. Existing methods (Preston et al., 2011) focus on one aspect of climate change (such as the expected economic loss (Burke et al., 2018)) using tailor-designed analysis, while H-divergence provides a general tool for hypothesis testing and visualization for different aspects of climate change. In our example, we choose suitable loss functions to quantitatively measure aspects of climate change that are relevant to decision making in agriculture and renewable energy production.[2]

**Setup** We use the NOAA database which contains daily weather from thousands of weather stations at different geographical locations. For each location, we summarize the weather sequence of each year into a few summary statistics (average yearly temperature, humidity, wind speed and rainy days). We are interested in assessing changes in weather over this period at each location, from the perspective of agriculture and renewable energy activities. Further details of these experiments are in Appendix C.2.

**Example: Agriculture** It is known that climate changes affect crop suitability (Lobell et al., 2008). Let $\mathcal{A}$ denote the set of possible crops to plant at each location (e.g. wheat/barley/rice), and $\ell(x, a)$ denote the loss of planting crop $a$ if the yearly weather is $x$. We estimate the function $\ell$ by matching geographical locations in the FAO crop yield dataset (FAOSTAT et al., 2006) to weather stations in the NOAA database, and learn a function to predict crop yield from weather data with kernel ridge regression.

The H-divergence has a natural interpretation: a geographical location could either (1) plant the same crop for the entire period 1981-2019 that is optimal for the local climate (i.e. choose $a^* = \arg\min_{a \in \mathcal{A}} \mathbb{E}_{(p+q)/2}[\ell(X, a)]$); (2) plant the optimal crops for 1981-1999 and for 2000-2019 respectively. H divergence measures the additional loss of option (1) compared to option (2). In other words, it is the excess loss of not adapting crop type to climate change. For each geographical location we can compute the H-divergence $D_\ell^{\text{JS}}$ for the estimated $\ell$ (plotted in Figure 3 left).

**Example: Energy production** Changes in weather also affect electricity generation, since climate change could affect the amount of wind/solar energy available. Let $\mathcal{A}$ denote the number of wind/solar/fossil fuel power plants built, and $\ell(x, a)$ denote the loss (negative utility) when the weather is $x$. We obtain the function $\ell$ using empirical formulas for energy production (Npower, 2012). The H-divergence for this loss function is shown in Figure 3 (right). Intuitively the H di-

---

[2]Designing loss functions $\ell$ that capture the effect of climate on human activities is a well studied topic in economics, and beyond the scope of this work. Our results should be taken as an illustrative example of how domain experts might use H-divergence with more realistic loss functions.

| Loss Selection | Selected Features |
|---|---|
| Neutral | education, cap-gain, sex, age, occupation |
| Upweight low income | education, cap-gain, relationship, marital-status, sex |
| Upweight high income | education, cap-gain, sex, age, race |

Table 2: Features selected by different approaches. With H-Divergence we can select different features that are important in different decision problems. For example, if we assign a high / low penalty to making incorrect prediction for higher income groups, we select a different set of features.

vergence measures the excess loss of using the same energy generation infrastructure for the entire time period vs. using different infrastructure that adapts to climate change. While this is only an illustrative example, comparing the two maps we see that regions and industries are affected by climate change in different ways – H divergence provides a quantitative framework for this kind of assessments.

## 5.2 FEATURE SELECTION

In a feature selection task, we wish to know which input features are most predictive of the label. Feature selection provides information on which features have the biggest influence on the label, and can be used in scientific discovery (Jović et al., 2015; Zhang et al., 2015).

Off-the-shelf feature selection algorithms often do not take into account problem specific requirements. For example, denote the input features as $X_1, \cdots, X_K$ and label as $Y$, the mutual information feature selection algorithms estimate the Shannon mutual information $I(X_i, Y) := \mathrm{KL}(p(X_i, Y) \| p(X_i) p(Y))$ and select features with largest mutual information. However, scientists and policies makers often need fine-grained control to answer their specific scientific or policy questions. For example, social scientists might want to know which features are more important for high-income as compared to low-income groups (e.g. to understand potential glass ceilings).

With H-Divergence we can select features with large $D_\ell^\phi(p(X_i, Y) \| p(X_i) p(Y))$ (i.e. the optimal action is different under the joint $p(X_i, Y)$ and the product of marginals $p(X_i) p(Y)$). By choosing different loss functions $\ell$ we can get different feature selection results, each reflecting important features for that decision problem. For example, in Table 2 we show the features selected for the UCI income prediction dataset (Blake, 1998). For this dataset, we choose $\mathcal{A}$ as the set of logistic regression functions, $l(x, a)$ as the cross entropy loss for regression function $a$ on the sample $x$ and $\phi(\theta, \lambda) = \max(\theta, \lambda)$. If we want to focus on high income groups, we can assign a higher weight to the loss of high income samples, and vice versa. We observe that gender/race is more predictive of income for high-income groups, while relationship or marital status is more predictive of income for lower-income groups. This can help us identify potential inequality or suggest further investigation into the cause of low income and poverty. For example, our results suggest a connection between family and relationship status and poverty, and a connection between gender/race and high income. These connections merit further investigation into the cause and policy remedy.

## 6 ACKNOWLEDGEMENTS

SE acknowledges support by NSF(#1651565, #1522054, #1733686), ONR (N000141912145), AFOSR (FA95501910024), ARO (W911NF-21-1-0125) and Sloan Fellowship.

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

# A  PROOFS

**Lemma 2.** *For any choice of $\ell$ and for any choice of $\phi$ that satisfy Definition 2, $D_\ell^\phi$ is non-negative and $D_\ell^\phi(p, q) = 0$ whenever $p = q$. Furthermore, $D_\ell^\phi$ is symmetric whenever $\phi$ is symmetric.*

*Proof of Lemma 2.* For any choice of $p, q$ by the concavity of the H-entropy in Lemma 1 we have

$$H_\ell \left( \frac{p + q}{2} \right) - H_\ell(p) \geq \frac{1}{2}(H_\ell(q) - H_\ell(p))$$

$$H_\ell \left( \frac{p + q}{2} \right) - H_\ell(q) \geq \frac{1}{2}(H_\ell(p) - H_\ell(q))$$

Therefore by summing the two inequalities we have

$$\left( H_\ell \left( \frac{p + q}{2} \right) - H_\ell(p) \right) + \left( H_\ell \left( \frac{p + q}{2} \right) - H_\ell(q) \right) \geq 0$$

By the requirement on $\phi$ we know that $\mathcal{D}_\ell^\phi(p \| q) \geq 0$. In addition when $p = q$ since $(p + q)/2 = p = q$ we have $D_\ell^\phi(p \| q) = \phi(0, 0) = 0$.

To show it is symmetric, note that

$$D_\ell^\phi(p \| q) = \phi \left( H_\ell \left( \frac{p + q}{2} \right) - H_\ell(p), H_\ell \left( \frac{p + q}{2} \right) - H_\ell(q) \right) = \phi \left( H_\ell \left( \frac{p + q}{2} \right) - H_\ell(q), H_\ell \left( \frac{p + q}{2} \right) - H_\ell(p) \right)$$

$$= D_\ell^\phi(q \| p)$$

whenever $\phi$ is symmetric.

$\square$

**Proposition 3.** *$D_\ell^\phi(p \| q) > 0$ if and only if $\arg\inf_a \mathbb{E}_p[\ell(X, a)] \cap \arg\inf_a \mathbb{E}_q[\ell(X, a)] = \emptyset$.*

*Proof of Proposition 3.* Denote $\mathcal{A}_p^* = \arg\inf_a \mathbb{E}_p[\ell(X, a)]$ and $\mathcal{A}_q^* = \arg\inf_a \mathbb{E}_q[\ell(X, a)]$. Also compute

$$H_\ell \left( \frac{p + q}{2} \right) = \inf_a \mathbb{E}_{\frac{p+q}{2}}[\ell(X, a)] = \inf_a \left( \frac{1}{2}\mathbb{E}_p[\ell(X, a)] + \frac{1}{2}\mathbb{E}_q[\ell(X, a)] \right) \tag{3}$$

If $\mathcal{A}_p^* \cap \mathcal{A}_q^* = \emptyset$, for any action $a'$ such that $\mathbb{E}_p[\ell(X, a')] = H_\ell(p)$, we must have $a' \in \mathcal{A}_p^*$ so $a' \notin \mathcal{A}_q^*$ and $\mathbb{E}_q[\ell(X, a')] > H_\ell(q)$. Similar if we choose $a''$ such that $\mathbb{E}_q[\ell(X, a'')] = H_\ell(q)$ we have similarly have $\mathbb{E}_p[\ell(X, a'')] > H_\ell(p)$. In other words, for any choice of action $a \in \mathcal{A}$ either $a \notin \mathcal{A}_p^*$ and $\mathbb{E}_p[l(X, a)] > H_\ell(p)$ or $a \in \mathcal{A}_p^*$ and $\mathbb{E}_q[l(X, a)] > H_\ell(q)$. Therefore

$$\inf_a \left( \frac{1}{2}\mathbb{E}_p[\ell(X, a)] + \frac{1}{2}\mathbb{E}_q[\ell(X, a)] \right) > \frac{1}{2}H_\ell(p) + \frac{1}{2}H_\ell(q) \tag{4}$$

Combining Eq.(3) and Eq.(4) we have

$$\frac{1}{2} \left( H_\ell \left( \frac{p + q}{2} \right) - H_\ell(p) \right) + \frac{1}{2} \left( H_\ell \left( \frac{p + q}{2} \right) - H_\ell(q) \right) > 0$$

By Definition 2 this implies (for any choice of $\phi$ that satisfies the requirements in Definition 2) that $D_\ell^\phi(p \| q) > 0$.

To prove the converse simply obverse that if $\mathcal{A}_p^* \cap \mathcal{A}_q^* \neq \phi$, let $a^* \in \mathcal{A}_p^* \cap \mathcal{A}_q^*$ we have $a^* = \arg\inf_{a \in \mathcal{A}} \mathbb{E}_{\frac{p+q}{2}}[l(X, a)]$. This implies that

$$2H_\ell \left( \frac{p + q}{2} \right) - H_\ell(q) - H_\ell(p) = 2\mathbb{E}_{\frac{p+q}{2}}[l(X, a^*)] - \mathbb{E}_q[l(X, a^*)] - \mathbb{E}_p[l(X, a^*)] = 0$$

By Definition 2 we can conclude that $D_\ell^\phi(p \| q) = 0$.

$\square$

**Theorem 1.** *The set of H-Jensen Shannon Divergences is strictly larger than the $\text{MMD}^2$ distances.*

*Proof of Theorem 1.* Let $k(x, y)$ be some kernel on an input space $\mathcal{X}$, and let $\mathcal{H}$ be the RKHS induced by the kernel. The (squared) MMD distance is defined by

$$\mathrm{MMD}^2(p, q) = \mathbb{E}_{X \sim p, Y \sim p} k(X, Y) + \mathbb{E}_{X \sim q, Y \sim q} k(X, Y) - 2 \mathbb{E}_{X \sim p, Y \sim q} k(X, Y)$$

which we write more compactly as $\mathrm{MMD}^2(p, q) = \mathbb{E}_{p,p} k(X, Y) + \mathbb{E}_{q,q} k(X, Y) - 2 \mathbb{E}_{p,q} k(X, Y)$.

Define $\phi(x, y) = \|k(x, \cdot) - k(y, \cdot)\|_{\mathcal{H}}^2$. We can rewrite this in the following form:

$$\mathrm{MMD}^2(p, q) = \mathbb{E}_{p,q} \phi(X, Y) - \frac{1}{2} \mathbb{E}_{p,p} \phi(X, Y) - \frac{1}{2} \mathbb{E}_{q,q} \phi(X, Y) \tag{5}$$

$$= \mathbb{E}_{p,q} \|k(X, \cdot)\|_{\mathcal{H}}^2 + \|k(Y, \cdot)\|_{\mathcal{H}}^2 - 2k(X, Y) - \frac{1}{2} \mathbb{E}_{p,p} \|k(X, \cdot)\|_{\mathcal{H}}^2 + \|k(Y, \cdot)\|_{\mathcal{H}}^2 - 2k(X, Y)$$

$$- \frac{1}{2} \mathbb{E}_{q,q} \|k(X, \cdot)\|_{\mathcal{H}}^2 + \|k(Y, \cdot)\|_{\mathcal{H}}^2 - 2k(X, Y) = \mathbb{E}_{p,p} k(X, Y) + \mathbb{E}_{q,q} k(X, Y) - 2 \mathbb{E}_{p,q} k(X, Y)$$

We also observe an algebraic relationship for any function $f(x, y)$ such that $f(x, y) = f(y, x)$ for all $x, y$:

$$\mathbb{E}_{\frac{p+q}{2}, \frac{p+q}{2}} f(X, Y) = \frac{1}{4} \mathbb{E}_{p,p} f(X, Y) + \frac{1}{4} \mathbb{E}_{p,q} f(X, Y) + \frac{1}{4} \mathbb{E}_{q,p} f(X, Y) + \frac{1}{4} \mathbb{E}_{q,q} f(X, Y)$$

$$= \frac{1}{4} \mathbb{E}_{p,p} f(X, Y) + \frac{1}{4} \mathbb{E}_{p,q} f(X, Y) + \frac{1}{4} \mathbb{E}_{q,p} f(Y, X) + \frac{1}{4} \mathbb{E}_{q,q} f(X, Y)$$

$$= \frac{1}{4} \mathbb{E}_{p,p} f(X, Y) + \frac{1}{4} \mathbb{E}_{q,q} f(X, Y) + \frac{1}{2} \mathbb{E}_{p,q} f(X, Y) \tag{6}$$

Furthermore, we have that

$$\mathbb{E}_{p,p} \|k(X, \cdot) - k(Y, \cdot)\|_{\mathcal{H}}^2 = 2 \mathbb{E}_p \|k(X, \cdot) - \mathbb{E}_p k(Y, \cdot)\|_{\mathcal{H}}^2 \tag{7}$$

Based on the above, noting that $\phi(x, y) = \phi(y, x)$, we can derive

$$\mathrm{MMD}^2(p, q) = \mathbb{E}_{p,q} \|k(X, \cdot) - k(Y, \cdot)\|_{\mathcal{H}}^2 - \frac{1}{2} \mathbb{E}_{p,p} \|k(X, \cdot) - k(Y, \cdot)\|_{\mathcal{H}}^2 - \frac{1}{2} \mathbb{E}_{q,q} \|k(X, \cdot) - k(Y, \cdot)\|_{\mathcal{H}}^2 \qquad \text{Eq (5)}$$

$$= 2 \mathbb{E}_{\frac{p+q}{2}, \frac{p+q}{2}} \|k(X, \cdot) - k(Y, \cdot)\|_{\mathcal{H}}^2 - \mathbb{E}_{p,p} \|k(X, \cdot) - k(Y, \cdot)\|_{\mathcal{H}}^2 - \mathbb{E}_{q,q} \|k(X, \cdot) - k(Y, \cdot)\|_{\mathcal{H}}^2 \qquad \text{Eq (6)}$$

$$= 4 \mathbb{E}_{\frac{p+q}{2}} \|k(X, \cdot) - \mathbb{E}_{\frac{p+q}{2}} k(Y, \cdot)\|_{\mathcal{H}}^2 - 2 \mathbb{E}_p \|k(X, \cdot) - \mathbb{E}_p k(Y, \cdot)\|_{\mathcal{H}}^2 - 2 \mathbb{E}_q \|k(X, \cdot) - \mathbb{E}_q k(Y, \cdot)\|_{\mathcal{H}}^2 \qquad \text{Eq (7)}$$

$$= 4 \inf_{a \in \mathcal{H}} \mathbb{E}_{\frac{p+q}{2}} \|k(X, \cdot) - a\|_{\mathcal{H}}^2 - 2 \inf_{a \in \mathcal{H}} \mathbb{E}_p \|k(X, \cdot) - a\|_{\mathcal{H}}^2 - 2 \inf_{a \in \mathcal{H}} \mathbb{E}_q \|k(X, \cdot) - a\|_{\mathcal{H}}^2. \qquad \text{mean def.}$$

Therefore we can define a loss $\ell : \mathcal{X} \times \mathcal{H} \to \mathbb{R}$ where

$$\ell(x, a) = 4 \|k(x, \cdot) - a\|_{\mathcal{H}}^2$$

Under the new notation we have

$$\mathrm{MMD}^2(p, q) = \inf_{a \in \mathcal{H}} \mathbb{E}_{\frac{p+q}{2}} l(X, a) - \frac{1}{2} \left( \inf_{a \in \mathcal{H}} \mathbb{E}_p l(X, a) + \inf_{a \in \mathcal{H}} \mathbb{E}_q l(X, a) \right)$$

$$= H_\ell \left( \frac{p+q}{2} \right) - \frac{1}{2} (H_\ell(p) + H_\ell(q)) = D_\ell^{\mathrm{JS}}(p \| q)$$

Conversely we want to show that not every H-Jensen Shannon divergence is a MMD. For example, take $H_\ell$ to be the Shannon entropy, then the corresponding $D_\ell^{\mathrm{JS}}$ is the Jensen-Shannon divergence, which is not a MMD. This is because the JS divergence is a type of $f$-divergence, and the only $f$-divergence that is also an IPM is total variation distance Sriperumbudur et al. (2009). Therefore, the set of H-Jensen Shannon Divergences is strictly larger than the set of MMDs. $\qquad \square$

**Theorem 2.** *If $\ell$ is C-bounded, and $\phi$ is 1-Lipschitz under the $\infty$-norm, for any choice of distribution $p, q \in \mathcal{P}(\mathcal{X})$ and $t > 0$ we have*

1. $\Pr[\hat{D}_\ell^\phi(\hat{p}_m \| \hat{q}_m) \geq t] \leq 4 e^{-\frac{t^2 m}{2C^2}}$ *if $p = q$.*
2. $\Pr \left[ \left| \hat{D}_\ell^\phi(\hat{p}_m \| \hat{q}_m) - D_\ell^\phi(p \| q) \right| \geq 4 \max(\mathcal{R}_m^p(\ell), \mathcal{R}_m^q(\ell)) + t \right] \leq 4 e^{-\frac{t^2 m}{2C^2}}$

*Proof of Theorem 2.* Let $\hat{p}_m$ be a sequence of $n$ samples $(x_1, \cdots, x_m)$ drawn from $p$, and $\hat{q}_m$ a sequence of $n$ samples $(x'_1, \cdots, x'_m)$ drawn from $q$. Let $\hat{r}_m$ the sub-sampling mixture $(x''_1, \cdots, x''_m)$ defined in Section 3.5 (i.e. $x''_i = x_i b_i + x'_i(1 - b_i)$ where $b_i$ is uniformly sampled from $\{0, 1\}$). We also overload the notation $H_\ell$ by defining $H_\ell(\hat{p}_m) = \inf_{a \in \mathcal{A}} \frac{1}{m} \sum_{i=1}^m l(x_i, a)$, and define $H_\ell(\hat{q}_m), H_\ell(\hat{r}_m)$ similarly.

Before proving this theorem we need the following Lemmas

**Lemma 3.** *Under the assumptions of Theorem 2*

$$\Pr\left[H_\ell(\hat{p}_m) - \mathbb{E}[H_\ell(\hat{p}_m)] \geq t\right] \leq e^{-\frac{2t^2 m}{C^2}}$$

$$\Pr\left[H_\ell(\hat{p}_m) - \mathbb{E}[H_\ell(\hat{p}_m)] \leq -t\right] \leq e^{-\frac{2t^2 m}{C^2}}$$

**Lemma 4.** *Under the assumptions of Theorem 2*

$$\Pr\left[|H_\ell(p) - H_\ell(\hat{p}_m)| \geq 2\mathcal{R}_m(\ell) + t\right] \leq e^{-\frac{2t^2 m}{C^2}}$$

To prove the first statement of the Theorem, when $p = q$ we can denote $\mu = \mathbb{E}[H_\ell(\hat{p}_m)] = \mathbb{E}[H_\ell(\hat{q}_m)] = \mathbb{E}[H_\ell(\hat{r}_m)]$, and we have

$$
\begin{aligned}
&\Pr\left[\hat{D}_\ell^\phi(\hat{p}_m \| \hat{q}_m) \geq t\right] \\
&= \Pr[\phi(H_\ell(\hat{r}_m) - H_\ell(\hat{p}_m), H_\ell(\hat{r}_m) - H_\ell(\hat{q}_m)) \geq t] && \text{Def 2} \\
&\leq \Pr\left[\max(H_\ell(\hat{r}_m) - H_\ell(\hat{p}_m), H_\ell(\hat{r}_m) - H_\ell(\hat{q}_m)) \geq t\right] && \phi \text{ 1-Lipschitz} \\
&\leq \Pr\left[H_\ell(\hat{r}_m) - H_\ell(\hat{p}_m) \geq t\right] + \Pr\left[H_\ell(\hat{r}_m) - H_\ell(\hat{q}_m) \geq t\right] && \text{Union bound} \\
&\leq \Pr\left[H_\ell(\hat{p}_m) - \mu \leq -t/2\right] + 2\Pr\left[H_\ell(\hat{r}_m) - \mu \geq t/2\right] + \Pr\left[H_\ell(\hat{q}_m) - \mu \leq -t/2\right] && \text{Union bound} \\
&\leq 4e^{-\frac{t^2}{2C^2/m}} && \text{Lemma 3}
\end{aligned}
$$

The third inequality is because if $H_\ell(\hat{r}_m) - H_\ell(\hat{p}_m) \geq t$ then it must be either $H_\ell(\hat{p}_m) - \mu \leq -t/2$ or $H_\ell(\hat{r}_m) - \mu \geq t/2$. Similarly if $H_\ell(\hat{r}_m) - H_\ell(\hat{q}_m) \geq t$ then it must be either $H_\ell(\hat{q}_m) - \mu \leq -t/2$ and $H_\ell(\hat{r}_m) - \mu \geq t/2$.

To prove the second statement of the Theorem, we observe that

$$
\begin{aligned}
&|\hat{D}_\ell^\phi(p_m \| q_m) - D_\ell^\phi(p \| q)| \\
&= \left|\phi\left(H_\ell(\hat{r}_m) - H_\ell(\hat{p}_m), H_\ell(\hat{r}_m) - H_\ell(\hat{q}_m)\right) - \phi\left(H_\ell\left(\frac{p+q}{2}\right) - H_\ell(p), H_\ell\left(\frac{p+q}{2}\right) - H_\ell(q)\right)\right| && \text{Def 2} \\
&\leq \max\left(\left|H_\ell(\hat{r}_m) - H_\ell(\hat{p}_m) - H_\ell\left(\frac{p+q}{2}\right) + H_\ell(p)\right|, \left|H_\ell(\hat{r}_m) - H_\ell(\hat{q}_m) - H_\ell\left(\frac{p+q}{2}\right) + H_\ell(q)\right|\right) && \phi \text{ 1-Lip} \\
&\leq \max\left(\left|H_\ell(\hat{r}_m) - H_\ell\left(\frac{p+q}{2}\right)\right| + |H_\ell(\hat{p}_m) - H_\ell(p)|, \left|H_\ell(\hat{r}_m) - H_\ell\left(\frac{p+q}{2}\right)\right| + |H_\ell(\hat{q}_m) - H_\ell(q)|\right) && \text{Jensen}
\end{aligned}
$$

Therefore, the event $|\hat{D}_\ell^\phi(p_m \| q_m) - D_\ell^\phi(p \| q)| \geq 4\max(\mathcal{R}_m^p(\ell), \mathcal{R}_m^q(\ell)) + t$ happens only if at least one of the following events happen

$$
\begin{aligned}
\left|H_\ell(\hat{r}_m) - H_\ell\left(\frac{p+q}{2}\right)\right| &\geq \mathcal{R}_m^p(\ell) + \mathcal{R}_m^q(\ell) + t/2 \geq 2\mathcal{R}_m^{(p+q)/2}(\ell) + t/2 && \mathcal{R} \text{ convex} \\
|H_\ell(\hat{p}_m) - H_\ell(p)| &\geq 2\mathcal{R}_m(\ell) + t/2 \\
|H_\ell(\hat{q}_m) - H_\ell(q)| &\geq 2\mathcal{R}_m(\ell) + t/2
\end{aligned}
$$

Based on Lemma 4 each of these events only happen with probability at most $e^{-\frac{t^2 m}{2C^2}}$. Therefore we can conclude by union bound that

$$\Pr[|\hat{D}_\ell^\phi(p \| q) - D_\ell^\phi(p \| q)| \geq 4\max(\mathcal{R}_m^p(\ell), \mathcal{R}_m^q(\ell)) + t] \leq 4e^{-\frac{t^2 m}{2C^2}}$$

Finally we prove the two Lemmas used in the theorem. Lemma 4 is a standard result in the Radamacher complexity literature. For a proof, see e.g. Section 26.1 in (Shalev-Shwartz & Ben-David, 2014). Lemma 3 can also be proved by standard techniques. We provide the proof here.

*Proof of Lemma 3.* Consider two sets of samples $x_1, \cdots, x_j, \cdots, x_m$ and $x'_1, \cdots, x'_j, \cdots, x'_m$ where $x_i = x'_i$ for every index $i = 1, \cdots, m$ except index $j$.

$$\left| \inf_a \frac{1}{m} \sum_i \ell(x_i, a) - \inf_a \frac{1}{m} \sum_i \ell(x'_i, a) \right| \leq \sup_a \left| \frac{1}{m} \sum_i \ell(x_i, a) - \frac{1}{m} \sum_i \ell(x'_i, a) \right|$$

$$= \frac{1}{m} \sup_a \left| \ell(x_j, a) - \ell(x'_j, a) \right| \leq \frac{C}{m}$$

Then we can conclude by Mcdiarmid inequality that

$$\Pr \left[ \inf_a \frac{1}{m} \sum_i \ell(X_i, a) - \mathbb{E} \left[ \inf_a \frac{1}{m} \sum_i \ell(X_i, a) \right] \geq t \right] \leq e^{-\frac{2t^2}{C^2/m}} = e^{-\frac{2t^2 m}{C^2}}$$

$\square$

$\square$

**Corollary 1.** $\sqrt{\mathrm{Var}[\hat{D}_\ell^\phi(\hat{p}_m \| \hat{q}_m)]} \leq 4 \max(\mathcal{R}_m^p(\ell), \mathcal{R}_m^q(\ell)) + \sqrt{2C^2/m}$

*Proof of Corollary 1.* For notation convenience denote $B = 4 \max(\mathcal{R}_m^p(\ell), \mathcal{R}_m^q(\ell))$

$$\mathrm{Var}[\hat{D}_\ell^\phi(\hat{p}_m \| \hat{q}_m)]$$

$$= \mathbb{E} \left[ \left( \hat{D}_\ell^\phi(\hat{p}_m \| \hat{q}_m) - D_\ell^\phi(p \| q) \right)^2 \right]$$

$$= \int_{t=0}^\infty \Pr \left[ \left( \hat{D}_\ell^\phi(\hat{p}_m \| \hat{q}_m) - D_\ell^\phi(p \| q) \right)^2 \geq t \right] dt$$

$$= \int_{t=0}^\infty \Pr \left[ \left| \hat{D}_\ell^\phi(\hat{p}_m \| \hat{q}_m) - D_\ell^\phi(p \| q) \right| \geq \sqrt{t} \right] dt$$

$$= \int_{s=0}^\infty \Pr \left[ \left| \hat{D}_\ell^\phi(\hat{p}_m \| \hat{q}_m) - D_\ell^\phi(p \| q) \right| \geq s \right] 2s \, ds \qquad\qquad s = \sqrt{t}$$

$$\leq \int_{s=0}^B 2s \, ds + \int_{s=0}^\infty \Pr \left[ \left| \hat{D}_\ell^\phi(\hat{p}_m \| \hat{q}_m) - D_\ell^\phi(p \| q) \right| \geq B + s \right] 2(B+s) \, ds$$

$$\leq B^2 + \int_{s=0}^\infty 2(B+s) e^{-\frac{s^2 m}{2C^2}} \, ds$$

$$\leq B^2 + \int_{t=0}^\infty 2(B + t\sqrt{\frac{2C^2}{m}}) e^{-t^2} \sqrt{\frac{2C^2}{m}} \, dt \qquad\qquad t = s\sqrt{\frac{m}{2C^2}}$$

$$= B^2 + 2B\sqrt{\frac{2C^2}{m}} \int e^{-t^2} dt + \frac{4C^2}{m} \int t e^{-t^2} dt$$

$$= B^2 + B\sqrt{\frac{2\pi C^2}{m}} + \frac{2C^2}{m} \leq (B + \sqrt{2C^2/m})^2$$

$\square$

**Corollary 2.** *[Consistency] Under the condition of Theorem 2, if additionally either 1. $\mathcal{A}$ is a finite set 2. $\mathcal{A}$ is a bounded subset of $\mathbb{R}^d$ for some $d \in \mathbb{N}$ and $\ell$ is Lipschitz w.r.t. $a$, then almost surely $\lim_{m \to \infty} \hat{D}_\ell^\phi(\hat{p}_m \| \hat{q}_m) = D_\ell^\phi(p \| q)$.*

*Proof of Corollary 2.* We can prove the consistency results from Theorem 2 by observing that the expected Radamacher complexity goes to 0 when $m \to \infty$.

The first statement is a simple consequence of Massart's Lemma, (e.g. see Eq.(8.44) in (Ma, 2021)). In particular, because $\mathcal{A}$ is finite we have

$$\mathcal{R}_m^p(\ell) \leq \sqrt{2 \log |\mathcal{A}|/m} \to_{m \to \infty} 0$$

To prove the second statement, first observe that because $\mathcal{A}$ is bounded, there must exist some $r \in \mathbb{R}$ such that $\mathcal{A} \subset B_r := \{a, \|a\|_2 \leq r\}$. In addition, without loss of generality we can assume that there exists $L \in \mathbb{R}$ such that $\forall x \in \mathcal{X}$ and $a, a' \in \mathcal{A}$

$$|\ell(x, a) - \ell(x, a')| \leq L\|a - a'\|_2$$

There is no loss of generality because in finite dimensions all norms are equivalent, so if $f$ is Lipschitz under any norm, then $\ell$ is Lipschitz under the 2-norm. We can apply the results on Radamacher complexity for smoothly parameterized class proved in (Bartlett, 2013), and conclude that $\lim_{m \to \infty} \mathcal{R}_m^p(\ell) = 0$. □

# B  ADDITIONAL EXPERIMENTAL RESULTS

## B.1  BLOB DATASET

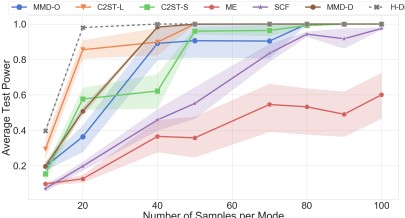
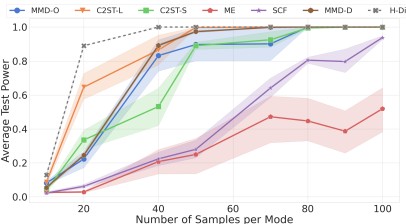

Figure 4: **Left:** Average test power on the Blob dataset for different sample sizes and significance level $\alpha = 0.05$. Our method (H-Div, dashed line) has significantly better test power, especially for setups with small sample sizes. **Right:** The same plot with significance level $\alpha = 0.01$.

## B.2  EVALUATING SAMPLE QUALITY

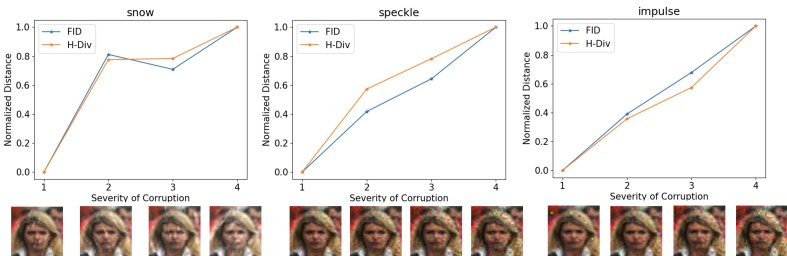

Figure 5: The divergence between corrupted image and original image measured by H-divergence vs. FID. For better comparison we normalize each distance to between $[0, 1]$ by a linear transformation. For "speckle" and "impulse" corruption, both divergences are monotonically increasing with more corruption. For "snow" corruption H-divergence is monotonic while FID is not. Other types of corruptions are provided in Appendix B.2.

The gold standard for evaluating image generative models requires human decision, which is nevertheless expensive. Several surrogate measurements are commonly used, such as the Frechet Inception Distance (FID) (Heusel et al., 2017) or the inception score. Here by formulating such evaluation as an estimation of the discrepancy between the generated and the real images, we can quantify the quality of image samples by calculating the corresponding H-Divergences.

We choose $\mathcal{A}$ as the set of Gaussian mixture distributions on the inception feature space, $l(x, a)$ as the negative log likelihood of $x$ under distribution $a$ and $\phi(\theta, \lambda) = \max(\theta, \lambda)$. To evaluate the performance, we use the same setup as (Heusel et al., 2017), where we add corruption from (Hendrycks & Dietterich, 2019) to a set of images. Intuitively, adding more corruption degrades the sample quality, so a good measurement of sample quality should assign a lower quality score (higher divergence from clean images). The results are plotted in Figure 5. The remaining plots of other perturbations are in Appendix B.2. Both FID and H-divergence are generally monotonically increasing as we increase the amount of corruption. Our method is slightly better on some perturbations (such as "snow"), where the FID fails to be monotonically increasing, but our method is still monotonic, better aligning with human intuition.

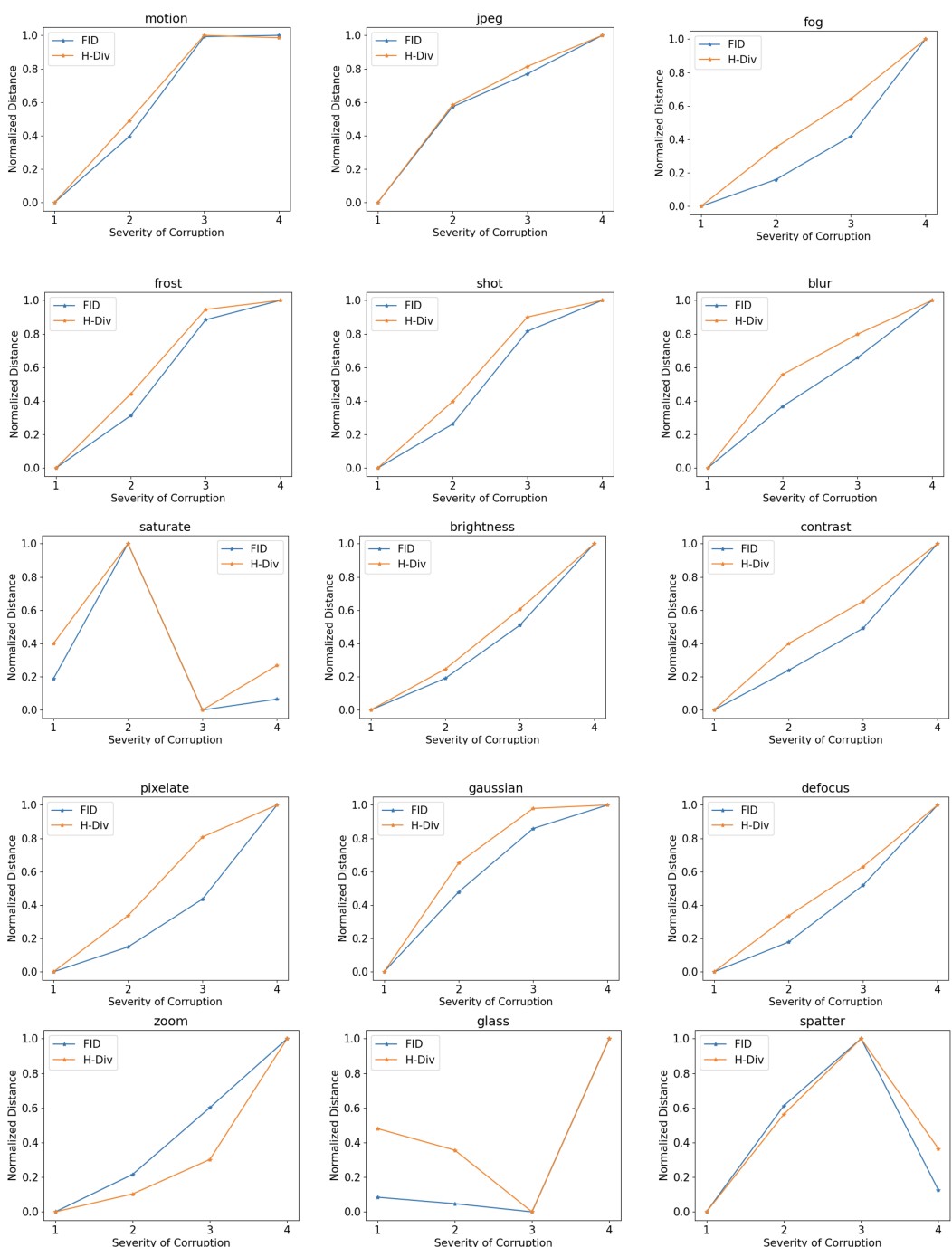

Figure 6: Additional plots that extend Figure 5.

# C ADDITIONAL THEORY AND EXPERIMENT DETAILS

## C.1 CONNECTION TO OPTIMAL TRANSPORT

We first show that H-divergence can also have a transportation interpretation. For all the intuitive interpretations we avoid technical difficulty by assuming $\mathcal{X}$ is a finite set, even though all the formulas are applicable when $\mathcal{X}$ has infinite cardinality.

| $N$ | ME | SCF | C2ST-S | C2ST-L | MMD-O | MMD-D | H-Div |
|---|---|---|---|---|---|---|---|
| 200 | $0.414\pm_{0.050}$ | $0.107\pm_{0.018}$ | $0.193\pm_{0.037}$ | $0.234\pm_{0.031}$ | $0.188\pm_{0.010}$ | $0.555\pm_{0.044}$ | $\mathbf{1.000}\pm_{\mathbf{0.000}}$ |
| 400 | $0.921\pm_{0.032}$ | $0.152\pm_{0.021}$ | $0.646\pm_{0.039}$ | $0.706\pm_{0.047}$ | $0.363\pm_{0.017}$ | $0.996\pm_{0.004}$ | $\mathbf{1.000}\pm_{\mathbf{0.000}}$ |
| 600 | $1.000\pm_{0.000}$ | $0.294\pm_{0.008}$ | $1.000\pm_{0.000}$ | $0.977\pm_{0.012}$ | $0.619\pm_{0.021}$ | $1.000\pm_{0.000}$ | $\mathbf{1.000}\pm_{\mathbf{0.000}}$ |
| 800 | $1.000\pm_{0.000}$ | $0.317\pm_{0.017}$ | $1.000\pm_{0.000}$ | $1.000\pm_{0.000}$ | $0.797\pm_{0.015}$ | $1.000\pm_{0.000}$ | $\mathbf{1.000}\pm_{\mathbf{0.000}}$ |
| 1000 | $1.000\pm_{0.000}$ | $0.346\pm_{0.019}$ | $1.000\pm_{0.000}$ | $1.000\pm_{0.000}$ | $0.894\pm_{0.016}$ | $1.000\pm_{0.000}$ | $\mathbf{1.000}\pm_{\mathbf{0.000}}$ |
| Avg. | 0.867 | 0.243 | 0.768 | 0.783 | 0.572 | 0.910 | **1.000** |

Table 3: Average test power $\pm$ standard error for $N$ samples over the MNIST dataset.

**Setup** Choose $\mathcal{A} = \mathcal{X}$, and let $l(x, a)$ be a symmetric function ($l(x, a) = l(a, x)$) that denotes the cost of transporting a unit of goods from $x$ to $a$. When we say that a unit of goods is located according to $p$, we mean that there is 1 unit of goods dispersed in $\mathcal{X}$ locations, where $p(x)$ is the amount of goods at location $x$.

**Optimal Transport Distance** The optimal transport distance is defined by

$$O_\ell(p, q) = \inf_{r_{XY}, r_X = p_X, r_Y = q_Y} \mathbb{E}_{r_{XY}}[l(X, Y)]$$

Intuitively the optimal transport distance measures the following cost: initially the goods are located according to $p$, we would like to move them to locate according to $q$; $O(p, q)$ denote the minimum cost to accomplish this transportation task.

**H-Divergence as Optimal Storage** We first consider the intuitive interpretation to the H-entropy

$$H_\ell(p) = \inf_a \mathbb{E}_p[\ell(X, a)] \qquad a^* = \arg\inf_{a \in \mathcal{X}} \mathbb{E}_p[\ell(X, a)]$$

Suppose we want to move goods located according to $p$ to a storage location (for example, we want to collect all the mail in a city to a package center), then $a^*$ is the optimal location to build the storage location, and H-entropy measures the minimum cost to transport all goods to the storage location. Similarly $2H_\ell\left(\frac{p+q}{2}\right)$ measures the minimum cost to transport both goods located according to $p$ and goods located according to $q$ to the same storage location. The H-divergence

$$2D_\ell^{\text{JS}}(p\|q) := 2H_\ell\left(\frac{p+q}{2}\right) - (H_\ell(q) + H_\ell(p))$$

measures the reduction of transportation cost with two storage locations (one for $p$ and one for $q$) rather than a single storage location (for both $p$ and $q$).

The H-Divergence is related to the optimal transport distance by the following inequality.

**Proposition 4.** *If $\ell$ satisfies the triangle inequality $\forall x, y, z \in \mathcal{X}, l(x, y) + l(y, z) \geq l(x, z)$ then $D_\ell^{\text{JS}}(p\|q) \leq \frac{1}{2}O(p, q)$*

*Proof of Proposition 4.* Let $a_q^* = \arg\inf \mathbb{E}_q[l(X, a)]$ then we have

$$\begin{aligned}
2H_\ell\left(\frac{p+q}{2}\right) &= \inf_a \left(\mathbb{E}_p[\ell(X, a)] + \mathbb{E}_q[\ell(X, a)]\right) \leq \mathbb{E}_p[\ell(X, a_q^*)] + \mathbb{E}_q[\ell(X, a_q^*)] \\
&\leq \inf_{r_{XY}, r_X = p_X, r_Y = q_Y} \mathbb{E}_{r_{XY}}[\ell(X, a_q^*)] + \mathbb{E}_q[\ell(X, a_q^*)] \\
&\leq \inf_{r_{XY}, r_X = p_X, r_Y = q_Y} \mathbb{E}_{r_{XY}}[\ell(X, Y) + \ell(Y, a_q^*)] + \mathbb{E}_q[\ell(X, a_q^*)] \\
&= O_\ell(p, q) + 2H_\ell(q)
\end{aligned}$$

Intuitively, to move goods located according to $p$ and goods located according to $q$ to some storage location, one option is to first transport all goods from $p$ to $q$ (so that the goods at location $x$ will be $2q(x)$), then move the goods located according to $2q$ to the optimal storage location. Similarly we have

$$2H_\ell\left(\frac{p+q}{2}\right) \leq O(q, p) + 2H_\ell(p)$$

which combined we have

$$2D_\ell^{\text{JS}}(p\|q) = 2H_\ell\left(\frac{p+q}{2}\right) - (H_\ell(q) + H_\ell(p)) \leq O(p, q)$$

$\square$

## C.2 IMPOSSIBILITY OF JENSEN-SHANNON DIVERGENCE ESTIMATION

Suppose we have a consistent estimator for the Jenson-Shannon divergence, i.e. a function $\hat{\text{JS}}$ such that for any pair of distribution $p, q$ and given $N$ i.i.d. samples $p_N \sim p$ and $q_N \sim q$ we have $\lim_{N\to\infty} \hat{\text{JS}}(p_N\|q_N) = \hat{\text{JS}}(p\|q)$, then we prove a contradiction by the probabilistic method.

Let $p$ be a standard Gaussian distribution, let $Q^M$ be a uniform distribution on a set of $M$ i.i.d. samples from $p$ (hence $Q^M$ is itself a random variable that depends on the i.i.d. samples). Let $Q^*$ be the limit of $Q^M$ when $M \to \infty$ (i.e. it is the uniform distribution on an infinite set of samples). Let $q^*$ denote a value that $Q^*$ can take. Because $q^*$ is always supported on countably many points, hence $\text{JS}(p\|q^*) = 1$. Note that for any $N$ the following two sampling process leads to identical distribution on $p_N, q_N$:

$$p_N \sim p, q_N \sim p \qquad Q^* \sim p, p_N \sim p, q_N \sim Q^*$$

Hence, the expectation of any function (including $\hat{\text{JS}}$) is also identical.

$$\mathbb{E}_{Q^*\sim p}\left[\mathbb{E}_{p_N\sim p, q_N\sim Q^*}[\hat{\text{JS}}(p_N, q_N)]\right] = \mathbb{E}_{p_N\sim p, q_N\sim p}[\hat{\text{JS}}(p_N, q_N)]$$

Hence the limit when $N \to \infty$ must be identical

$$\lim_{N\to\infty} \mathbb{E}_{Q^*\sim p}\left[\mathbb{E}_{p_N\sim p, q_N\sim Q^*}[\hat{\text{JS}}(p_N, q_N)]\right] = \lim_{N\to\infty} \mathbb{E}_{p_N\sim p, q_N\sim p}[\hat{\text{JS}}(p_N, q_N)]$$

Because the Jenson Shannon divergence is always bounded, any consistent estimator must also be bounded for sufficiently large $N$. By the dominated convergence theorem we can exchange the expectation and the limit.

$$\mathbb{E}_{Q^*\sim p}\left[\lim_{N\to\infty} \mathbb{E}_{p_N\sim p, q_N\sim Q^*}[\hat{\text{JS}}(p_N, q_N)]\right] = \lim_{N\to\infty} \mathbb{E}_{p_N\sim p, q_N\sim p}[\hat{\text{JS}}(p_N, q_N)]$$

By the probabilistic method (i.e. for any function $f$ if $\mathbb{E}_{Q^*\sim p}[f(Q^*)] = 0$ there must exist some $q^*$ such that $f(q^*) \leq 0$) there must exist some $q^*$ such that

$$\lim_{N\to\infty} \mathbb{E}_{p_N\sim p, q_N\sim q^*}[\hat{\text{JS}}(p_N, q_N)] \leq \lim_{N\to\infty} \mathbb{E}_{p_N\sim p, q_N\sim p}[\hat{\text{JS}}(p_N, q_N)] = 0 \neq \text{JS}(p\|q^*) = 1$$

Therefore $\hat{\text{JS}}$ cannot be consistent.

## C.3 CLIMATE CHANGE EXPERIMENT DETAILS

**Setup Details** In this experiment, we extract the statistics of yearly weather for each year from 1981-2019. We use the NOAA dataset, which contains daily weather data from thousands of weather stations across the globe. For each year we compute the following summary statistics: average yearly temperature, average yearly humidity, average yearly wind speed and average number of rainy days in an year. For example $x_{1990}$ is a 4 dimensional vector where each dimension correspond to one of the summary statistics above.

Let $p$ denote the uniform distribution over $\{x_{1981}, \cdots, x_{1999}\}$ and $q$ denote the uniform distribution over $\{x_{2000}, \cdots, x_{2019}\}$. For example $\mathbb{E}_p[\ell(X, a)]$ denote the expected loss of action $a$ for a random year sampled from 1981-1999. Note that for many decision problems, it is possible to make yearly decisions (e.g. decide the best crop to plant for each year). However, because we want to measure the difference between two time periods 1981-1999 vs. 2000-2019, we choose the action space $\mathcal{A}$ to be a single crop selection that will be used for the entire time period (rather than a different crop selection for each year). Similarly for energy production we choose the action space $\mathcal{A}$ to be the proportion of different energy production methods that will be used for the entire time period.

**Crop yield** We obtain the crop yield loss function $\ell(x, a)$ with the following procedure

1. We obtain the crop yield dataset from (FAOSTAT et al., 2006), each entry we extract is the following tuple: (country code, year, crop type, yield per hectare (kg/ha))
2. We associate each country code with the central coordinate (i.e. the average latitude and longitude) of the country. For each central coordinate we find the nearest weather station in the NOAA database. We use data for the nearest weather station as the weather data for the country.
3. Based on step 2 for each (country code, year) pair we can associate a weather statistics (i.e. the 4 dimensional vector described in Setup Details). We update each entry in step 1 to be (weather statistics, crop type, yield per hectare).
4. Based on the data entries we obtain in step 3 we train a Kernel Ridge regression model to learn the function $\ell(x, a)$ where $x$ is the weather statistics, $a$ is the crop type, and $\ell(x, a)$ is learned to predict the yield (normalized by market price) of the weather $x$ for crop type $a$.

**Energy production** We consider three types of energy production methods: solar, wind and traditional (such as fossil fuel). Solar energy and wind energy both depend heavily on weather, while traditional energy does not. In particular, we use empirical formulas for solar and wind energy calculation:

$$\text{solar} \propto \text{number of sunny days} * \text{daylight hour}$$

$$\text{wind} \propto \text{wind velocity}^3$$

## D  DISCUSSIONS

**Future Work** In this paper we explored the applications of H-divergence to two sample testing. Future work can explore other applications of divergences. Potential applications include

- Generative model training. Many generative models learning algorithm minimize divergences (Nowozin et al., 2016; Arjovsky et al., 2017), and future work can explore if the new divergence family leads to new generative model learning algorithms.
- Independence tests. Independence tests are two sample tests between the joint distribution $p_{XY}$ and the product of marginal distributions $p_X p_Y$, therefore the two sample test results from this paper are applicable to independence tests.
- Robustness. Many robust optimization, estimation or prediction methods aim to achieve good performance even when the data distribution is perturbed. Typically perturbation is measured by e.g. the KL divergence or the Lp distances. Future work can measure perturbation with the H-divergence $H_\ell$ by choosing loss functions $\ell$ that are tailored for the problem.

**Computation Issues** There are several situations where estimating the H-divergence is (provably) computationally feasible:

- When $\mathcal{A}$ is a small finite set, in which case we can enumerate all possible values of $a \in \mathcal{A}$.
- When the loss function $\ell(y, a)$ is convex in $a$, in which case we can accurately estimate the H-divergence in polynomial time by solving the optimization problem $\inf_a \mathbb{E}[\ell(Y, a)]$ by gradient descent.

In general, while it is difficult to guarantee computational feasibility, we use a practical technique that works well in our experiments: we use the same number of gradient descent steps for evaluating $H_\ell\left(\frac{p+q}{2}\right)$ and $H_\ell(p)$, $H_\ell(q)$. Intuitively, the "sub-optimality" when estimating the three terms is approximately the same in expectation and cancels out.

