# OpenReview forum: "Comparing Distributions by Measuring Differences that Affect Decision Making"
_ICLR.cc/2022/Conference — ICLR 2022 Oral_

### Official Review · Reviewer_SZbR · 2021-10-31

**Correctness:** 3
**Technical Novelty And Significance:** 3
**Empirical Novelty And Significance:** 3
**Recommendation:** 8
**Confidence:** 3

**Main Review:**

Strengths:

Proposes a new type of divergence that compares two probability distributions from the lens of optimal decision-making. By setting an action set and loss function, one can identify "how far two distributions are" in terms of leading to the same optimal action. Moreover, the paper clearly explains the connection of H-divergence with f-divergence and integral probability metrics. Overall, H-divergence can be seen as a valuable contribution to the family of divergences used in machine learning tasks.

Experiments are satisfactory. Especially, the flexibility achieved by using H-divergence by choosing application tailored actions and losses highlight the merit of H-divergence in performing two samples test.

Weaknesses:

The convergence properties of the empirical estimator of H-divergence deserve more discussion in the main paper. In particular, the role of Rademacher complexity in the bounds in Theorem 2 and Corollary 1 is not clearly explained. It will be good to add a discussion in line with proof of Corollary 1 (on how fast Rademacher complexity vanishes) into the main paper.

The paper does not give enough detail about computational aspects. This clearly depends on the structure of the action set and loss function. The paper would benefit from the characterization of a set of (general) sufficient conditions under which computation of H-divergence is feasible.


**Summary Of The Paper:**

This paper proposes H-divergence, a new type of divergence based on H-entropy, to compare two probability distributions. This new divergence includes some of the commonly used integral probability metrics and f-divergences, such as Jensen Shannon divergence and MMD as special cases. A crucial property of H-divergence is that it takes into account the decision loss; namely, it compares two distributions in a way that distinguishes them based on the optimal decision loss, i.e., "two distributions are different if the optimal decision loss is higher on their mixture than on each individual distribution." It also provides an empirical estimator for H-divergence and studies its convergence properties. The paper studies several use cases of H-divergence, including two-sample tests. Experiments demonstrate that H-divergence achieves higher test power than tests based on MMD under the same type I error rate. Another important use case of H-divergence is understanding whether differences between distributions are significant enough to affect decision making (from the viewpoint of minimizing loss) in different experiments, including how climate change affects various economic activities.


**Summary Of The Review:**

This paper makes a fundamental contribution to the family of divergences used for machine learning tasks and provides convincing evidence on the usefulness of H-divergence. However, there is still room for improvement related to the computation of H-divergence. Please describe the feasibility of computation of H-divergence for the experimental examples studied in the paper in your response.

---

> ### Author Response · Authors · 2021-11-23
> **Thank you for your review.**
>
> Thank you for your detailed review and suggestions!
>
> **Q: The convergence properties of the empirical estimator of H-divergence deserve more discussion in the main paper. In particular, the role of Rademacher complexity in the bounds in Theorem 2 and Corollary 1 is not clearly explained.**
>
> Thank you for this suggestion. We have added additional discussions in the updated paper. For both cases in Corollary 1, the Rademacher complexity vanishes at a rate of $O(1/\sqrt{m})$ when the sample size $m \to \infty$. Combined with Theorem 2, we conclude that the estimation error for H-divergence vanishes at a rate of $O(1/\sqrt{m})$.
>
> **Q: Need to discuss computational aspects. This clearly depends on the structure of the action set and loss function. The paper would benefit from the characterization of a set of (general) sufficient conditions under which computation of H-divergence is feasible.**
>
> We have added a discussion section in the revised draft (currently in the appendix due to length limitations, but will be moved to main paper whenever length permits):
>
> There are several situations where estimating the H-divergence is computationally feasible with guarantees:
>
> 1. When $\mathcal{A}$ is a small finite set, in which case we can enumerate all possible values of $a \in \mathcal{A}$.
> 2. When $\mathcal{A}$ is a $d$-dimensional vector space, and the loss function $\ell(y, a)$ is convex in $a$, in which case we can accurately estimate the H-divergence in polynomial time by solving the optimization problem $\inf_a  \mathbb{E}[\ell(Y, a)]$ with gradient descent.
>
> With more complex loss functions (such as when the action is a deep generative model), it can be difficult to guarantee exact H-divergence estimation under bounded computation. Nevertheless a practical technique works well in our experiments: we use the same number of gradient descent steps for evaluating $H_\ell\left(\frac{p+q}{2}\right)$, $H_\ell(p)$, and $H_\ell(q)$. Intuitively, any "optimization sub-optimality" when estimating the three terms is roughly the same and approximately cancels out.

---

### Official Review · Reviewer_jd8q · 2021-11-02

**Correctness:** 4
**Technical Novelty And Significance:** 4
**Empirical Novelty And Significance:** 3
**Recommendation:** 8
**Confidence:** 3

**Main Review:**

Strengths

1. Measuring the difference between two probability distributions is a fundamental problem in machine learning. Although many different types of divergencies have been proposed in the literature, they are typically decision independent. H-divergence seems a simple yet effective way of incorporating decision related domain knowledge into the discrepancy measure.
2. The theoretical results are non-trivial are provide insightful connections between H-divergence and other commonly used discrepancies.
3. The paper is well written. The theoretical results are solid and clearly explained. The three examples  clearly illustrate the broad applicability of H-divergence.

Questions:

1. Theorem 2 assumes that the same number of data points are sampled from p and q, respectively. I wonder if the result can be generalized to the case when the number of samples is different for p and q.
2. Can H-divergence still be useful when the decision task is initially unknown (as in reward-free reinforcement learning) or uncertain (as in multi-task learning)?
3. In Section 4.2, only the results for alpha = 0.05 are given. I wonder if similar patterns hold for other values of alpha.


**Summary Of The Paper:**

This paper proposes a new category of divergences, called H-divergence, for measuring the discrepancy  between two probability distributions. H-divergence is based on the optimal loss with respect to a chosen decision task and is therefore decision dependent. Further, it generalizes a few well known divergences such as the Jensen-Shannon divergence and the maximum mean discrepancy family. The paper proves a few properties of H-divergence including a convergence result when H-divergence is estimated using a finite set of samples. Further, three examples are used to illustrate the applications of H-divergence  including two sample tests, assessing climate change, and feature selection, which demonstrate the advantage of H-divergence compared with other commonly used discrepancies.

**Summary Of The Review:**

Overall I think this is a very interesting paper. The new family of discrepancies is likely to find broad applications in machine learning and data science and can potentially inspire the development of other decision dependent discrepancy measures.

---

> ### Author Response · Authors · 2021-11-23
> **Thank you for your review.**
>
> Thank you for your detailed review and suggestions!
>
> **Q: Theorem 2 assumes that the same number of data points are sampled from p and q, respectively. I wonder if the result can be generalized to the case when the number of samples is different for p and q.**
>
> It is straightforward to generalize Theorem 2.2 when the number of samples from p and q are different. This is because in the proof we separately bound the estimation error on the three terms $H_\ell(p)$, $H_\ell(q)$ and $H_\ell((p+q)/2)$, and each of the three bounds hold for any number of samples.
>
> Nevertheless, unless there is a big difference in the number of available samples, we recommend using the same number of samples for p, q. This is due to our insight for efficient estimation: if we use n samples to estimate each of the three terms: $H_\ell((p+q)/2)$ and $H_\ell(p)$, $H_\ell(q)$, they will approximately have the same amount of “overfitting” which cancel out. For example, when $p=q$ we prove a very low estimation error (Theorem 2.1) that is independent of the complexity of the loss function $\ell$.
>
> **Q: Can H-divergence still be useful when the decision task is initially unknown (as in reward-free reinforcement learning) or uncertain (as in multi-task learning)?**
>
> This is a great question, and in fact, we are currently working on extending H-divergence to multiple loss functions, where we define a new divergence that is the supremum of H-divergences for a family of loss functions. Intuitively, two distributions have positive divergence if there is at least one loss function with different optimal action. We find this construction useful in Bayesian optimization, robust optimization, and two-sample testing when the practitioner only knows the family of loss functions rather than an exact loss function.
>
> **Q: In Section 4.2, only the results for alpha = 0.05 are given. I wonder if similar patterns hold for other values of alpha.**
>
> We have experimented additionally on alpha=0.01 and updated the paper with these results in Figure 4 in the Appendix. The results are qualitatively similar: by reducing significance level, all methods have less test power, but H-divergence still achieves the best test power across the board.

---

### Official Review · Reviewer_pdf2 · 2021-11-04

**Correctness:** 4
**Technical Novelty And Significance:** 3
**Empirical Novelty And Significance:** 4
**Recommendation:** 8
**Confidence:** 3

**Details Of Ethics Concerns:**

I do not find any ethical issues with this paper.

**Main Review:**

**Strengths:**

(a) The proposed class of divergences, H-divergences, includes well-known divergences such as Jensen Shannon divergence and squared Maximum Mean Discrepancy as special cases. The H-divergence also contains new classes of divergences, including the ones defined in Equations (1) and (2), which appear to be potentially useful in practice.

(b) Some results are given to prove the convergence of the empirical H-divergence to its theoretical one; see Theorem 2 and Corollary 1.

(c) The conditions on the non-negativity of the H-divergence are obtained; see Section 3.3. These theoretical results partly justify the proposed divergence as a reasonable discrepancy between two probability distributions.

(d) A sufficient number of experiments are given to demonstrate the usefulness of the presented methods; see Sections 4 and 5.

(e) The paper is clearly written. Appendix provides helpful information including detailed proofs of the theoretical results of the main article.

&nbsp;

**Weaknesses:**

(f) Applications of divergences include not only two sample tests but also other methods such as robust estimation and independence tests. However the paper does not discuss possible applications other than two sample tests and plots in Figure 3. A brief discussion about other applications would be helpful for readers.

(g) The variance of the proposed estimator $\hat{D}^{\phi}_{\ell}  (\hat{p}_m || \hat{q}_m) $ is not discussed in the paper. Is it possible to discuss any result about the variance?

&nbsp;

**Minor Comments:**

(h) p.1, Section 1, 1st paragraph, l.13: Add a full stop after 'point'.

(i) p.1, Section 1, 2nd paragraph, l.1, etc.: There are at least three different expressions, namely, *H*-divergence, H-divergence and H-Divergence, in the paper to denote the same divergence. The expression should be standardized throughout the paper.

(j) p.4, Section 3.3, l.6 up: This property allow us to  ===>  This property allows us to

**Summary Of The Paper:**

This paper presents a new class of discrepancies between two continuous probability distributions. The proposed class, which is called the H-divergence, contains an extended class of Jensen Shannon divergences called H-Jensen Shannon divergences and another class (2) called the H-Min divergences as special cases. The conditions that the two probability distributions have non-negative H-divergence are given. It is seen that the set of H-Jensen Shannon divergences includes the set of squared Maximum Mean Discrepancies as a subset. Estimation and convergence of the H-divergence are discussed. The H-divergence is applied to propose two-sample tests and experiments suggest that the proposed tests outperform some existing tests in terms of power. The proposed methods are applied to climate data for decision making in agriculture and energy production.

**Summary Of The Review:**

The paper is generally well-written. Properties of the proposed class of divergences are well-investigated. A sufficient number of experiments are provided to demonstrate the usefulness of the proposed method.

---

> ### Author Response · Authors · 2021-11-23
> **Thank you for your review.**
>
> Thank you for the thoughtful reviews and suggestions!
>
> **Q: Applications of divergences include not only two-sample tests but also other methods such as robust estimation and independence tests. A brief discussion about other applications would be helpful for readers.**
>
> Thank you for this suggestion. We have added a discussion section (currently in the Appendix for space constraints but will be moved to the main paper whenever length permits).
>
> Specifically, independence tests are two sample tests between p(x, y) and p(x)p(y), hence it is natural to use our method for independence tests. We can also design new robust estimation or optimization procedures by measuring perturbation with H-divergence. We have suggested additional applications in the discussion section such as training generative models.
>
> **Q: The variance of the proposed estimator is not discussed in the paper. Is it possible to discuss any result about the variance?**
>
> Theorem 2 shows that the probability of large deviation is exponentially small. Therefore, the variance is bounded (in fact, all moments are bounded). In the revised paper, we added Corollary 1 to show that the standard deviation of the estimator is at most $4 \max(\mathcal{R}^p_m(\ell), \mathcal{R}^q_m(\ell)) + \sqrt{2C^2/m}$ where $m$ is the number of samples, and $\mathcal{R}_m$ is the Radamacher complexity as defined in Section 3.5 which usually goes to $0$ at a rate of $O(1/\sqrt{m})$.

---

> > ### Comment · Reviewer_pdf2 · 2021-11-29
> > **Re: Thank you for your review.**
> >
> > I would like to thank the authors for their careful responses to my comments.
> >
> > I have read the responses and confirmed the changes which had been made to the paper. I am satisfied with the new additions to the paper based on my comments (f) and (g). I will keep my good score.

---

### Decision · Program_Chairs · 2022-01-20

**Decision:**

Accept (Oral)

**Comment:**

Although by now there are several approaches for comparing probability distribution, the paper innovates by making their measure take into account the decision space and loss functions directly. The paper also frames its contribution within the literature at large. Reviewers were unanimous that the result is of major interest to the ICLR audience.